# Beam Theory of Thermal–Electro-Mechanical Coupling for Single-Wall Carbon Nanotubes

**DOI:** 10.3390/nano11040923

**Published:** 2021-04-05

**Authors:** Kun Huang, Ji Yao

**Affiliations:** 1Department of Engineering Mechanics, Faculty of Civil Engineering and Mechanics, Kunming University of Science and Technology, Kunming 650500, China; yaoji1974@163.com; 2Yunnan Key Laboratory of Disaster Reduction in Civil Engineering, Kunming University of Science and Technology, Kunming 650500, China

**Keywords:** single-walled carbon nanotubes, thermal–electro-mechanical coupling, Bernoulli–Euler beam theory, independent stiffness

## Abstract

The potential application field of single-walled carbon nanotubes (SWCNTs) is immense, due to their remarkable mechanical and electrical properties. However, their mechanical properties under combined physical fields have not attracted researchers’ attention. For the first time, the present paper proposes beam theory to model SWCNTs’ mechanical properties under combined temperature and electrostatic fields. Unlike the classical Bernoulli–Euler beam model, this new model has independent extensional stiffness and bending stiffness. Static bending, buckling, and nonlinear vibrations are investigated through the classical beam model and the new model. The results show that the classical beam model significantly underestimates the influence of temperature and electrostatic fields on the mechanical properties of SWCNTs because the model overestimates the bending stiffness. The results also suggest that it may be necessary to re-examine the accuracy of the classical beam model of SWCNTs.

## 1. Introduction

The excellent mechanical and electrical properties of carbon nanotubes (CNTs) have attracted significant attention from researchers. However, there is no widely accepted theory to model their mechanical properties [1,2,3,4,5]. In particular, the mechanical properties under combined physical fields (for example, temperature and electric field coupling) have not yet been thoroughly researched [5,6,7]. For the mechanical problems of single-walled carbon nanotubes (SWCNTs), there are two extensively discussed topics: first, the small-scale effect, i.e., the mechanical properties are dependent on the geometrical dimension [4,8,9,10]; second, how to determine the bending stiffness [3,11]. There is plenty of research on the small-scale effect, but these studies simultaneously introduce many controversies [4,9,12,13,14]. In most existing research, the classical beam model is modified through non-local theories to consider the small-scale effect [4,14,15]. These theories mainly include the stress gradient theory, $\left(1-a^{2}\nabla^{2}\right)\sigma_{ij}=E_{ijkl}\varepsilon_{kl}$, and the strain gradient theory, $\sigma_{ij}=C_{ijkl}\left(1-a^{2}\nabla^{2}\right)\varepsilon_{kl}$. Here, $\sigma_{ij}$ and $\varepsilon_{kl}$ are stress and strain, $E_{ijkl}$ and $C_{ijkl}$ are two four-rank tensors, a is the scale parameter. However, when applying non-local theories to CNTs, the question of how to determine the scale parameter a remains unanswered. Moreover, molecular dynamics (MD) simulations have revealed a bewildering problem: the scale parameter varies concerning length-to-diameter ratios, mode shapes, and boundary conditions [10]. The fact that the intrinsic material parameter, a, loses its uniqueness is odd. Fortunately, MD calculations also show that the small-scale effect can be ignored for SWCNTs with larger long-diameter ratios [10]. The discussion below only involves slender tubes, and the small-scale effect is ignored.

In fact, the subject of how to determine the bending stiffness of SWCNTs may be more important than the small-scale effect [3,11]. In the macro Bernoulli–Euler beam theory, the axial extensional stiffness and bending stiffness of beams are EA and EI. Here E, A, and I are the Young’s modulus, the cross-sectional area, and the cross-sectional moment of inertia, respectively. For a hollow circular section where A=πdh and $I=\pi \left(d^{3}h+dh^{3}\right)$, d and h are the diameter and the wall thickness, as shown in Figure 1. Obviously, for the classical beam theory, the extensional stiffness and bending stiffness are related to the tube’s wall thickness, but an SWCNT’s wall thickness is uncertain. This makes defining the bending stiffness by analogy with the classical beam theory difficult: the uncertainty of an SWCNT’s wall thickness leads to uncertainty regarding the bending stiffness. In fact, the uncertainty of SWCNTs’ bending stiffness results from the uncertainty of graphene’s bending stiffness. This is called the Yakobson paradox [8]. For graphene, the Yakobson paradox can be avoided by treating the bending stiffness and extensional stiffness independently [8,16,17]. Recent atomic calculations have shown that the bending stiffness and extensional stiffness of SWCNTs are also independent [3,11].

In application, CNTs are often in an environment with multiple coexisting fields, such as ambient heat and electrostatic and magnetic fields. However, the multifield coupling mechanical problems of CNTs still lack comprehensive research [6]. The present paper will focus on the influence of ambient temperatures and electrostatic fields on the mechanical properties of SWCNTs. For graphene, both experiments and atomic calculations show that it has a negative coefficient of thermal expansion (CTE) under finite ambient temperatures, i.e., graphene is a heat-shrinking and cold-expanding material [8,18]. However, the existing SWCNT research has shown that the radial CTE is negative, but the axial CTE is contradictory. Early MD calculations show that the axial CTE of SWCNTs is negative if the temperature is less than around 1000 K, but recent theories and experiments have confirmed that the axial CTE is positive [19,20,21]. For the effects of the electrostatic field on the mechanical properties, density functional theory (DFT) calculations and experiments show that the electrostatic field leads to the elongation of an SWCNT. The elongations induced by an electrostatic field or temperature may cause SWCNTs to buckle, and this problem has not yet been studied.

At the time of writing, there is no literature reporting the effects of independent bending and extensional stiffness on the vibrations of SWCNTs. Due to the experimental difficulty, there are no bending deformational experimental data for short SWCNTs (<30 nm) in the existing literature. The properties of an SWCNT’s thermal–electro-mechanical coupling have not been investigated by researchers. In this paper, we propose a beam model to take into account the above factors.

## 2. Thermal–Electro-Mechanical Coupling Beam Model with Two Independent Stiffnesses

Here, it is important to note that the sixfold symmetry of the graphene lattice is destroyed as graphene curls to form SWCNTs. This leads to significant anisotropy for the chiral (*m*,*n*). SWCNTs that have *m*≠*n*, but armchair (*n*,*n*) nanotubes are still isotropic [22,23,24,25,26,27]. For simplicity, this paper only considers (*n*,*n*) tubes, but the present model also can describe approximately other SWCNTs if their anisotropy is ignored. Neglecting entropy, the tube’s deformation energy includes bending energy, axis elongation deformation energy, thermal stress deformation energy, and electrostatic stress deformation energy. Therefore, the total deformation energy density is $F=F\left(\varepsilon_{xx}^{0},\kappa ,T,U\right)$. Here, $\varepsilon_{xx}^{0}$, κ, T and U are the axis elongation strain, bending curvature, absolute temperature, and electrostatic field intensity, respectively. According to Landau and Lifshitz’s expansion theory [28,29], the deformation energy density is expanded to the second order as
(1)$$\begin{array}{l} F=F_{0}+{\left(\frac{\partial F}{\partial \varepsilon_{xx}^{0}}\right)}_{\varepsilon_{xx}^{0}=0}\varepsilon_{xx}^{0}+{\left(\frac{\partial F}{\partial \kappa }\right)}_{\kappa =0}\kappa +{\left(\frac{\partial F}{\partial T}\right)}_{T=0}T+{\left(\frac{\partial F}{\partial U}\right)}_{U=0}U\\ +\frac{1}{2}k_{S}{\left(\varepsilon_{xx}^{0}\right)}^{2}+\frac{1}{2}k_{B}\kappa^{2}\kappa +k_{SB}\varepsilon_{xx}^{0}\kappa +k_{S}k_{T}\varepsilon_{xx}^{0}T+k_{S}k_{U}\varepsilon_{xx}^{0}U+\cdots . \end{array}$$

Here, $k_{S}={\left(\partial^{2}F/\partial {\left(\varepsilon_{xx}^{0}\right)}^{2}\right)}_{\varepsilon_{xx}^{0}=0}$ is the extensional stiffness; $k_{B}={\left(\partial^{2}F/\partial \kappa^{2}\right)}_{\kappa =0}$ is the bending stiffness; $k_{S}k_{U}={\left(\partial^{2}F/\partial \varepsilon_{xx}^{0}\partial U\right)}_{\varepsilon_{xx}^{0}=U=0}$, and $k_{U}$ is the coefficient of electrical expansion (CEE); $k_{S}k_{T}={\left(\partial^{2}F/\partial \varepsilon_{xx}^{0}\partial T\right)}_{\varepsilon_{xx}^{0}=T=0}$, and $k_{T}$ is the CTE; $k_{SB}={\left(\partial^{2}F/\partial \varepsilon_{xx}^{0}\partial \kappa \right)}_{\varepsilon_{xx}^{0}=\kappa =0}$ is the extensional bending stiffness, and $k_{SB}=0$ for (n,n) tubes [25,26,27]. Using this in the strain-free, curvature-free, and T=U=0 state, the axis stress and moment are zero, and we see from Equation (1) that the coefficients of the linear terms are zero, i.e., ${\left(\partial F/\partial \varepsilon_{xx}^{0}\right)}_{\varepsilon_{xx}^{0}=0}={\left(\partial F/\partial \kappa \right)}_{\kappa =0}={\left(\partial F/\partial T\right)}_{T=0}={\left(\partial F/\partial U\right)}_{U=0}=0$. Let $F_{0}=0$, so Equation (1) can be simplified as
(2)$$F=\frac{1}{2}k_{S}{\left(\varepsilon_{xx}^{0}\right)}^{2}+\frac{1}{2}k_{B}\kappa^{2}+k_{SB}\varepsilon_{xx}^{0}\kappa +k_{T}\varepsilon_{xx}^{0}T+k_{U}\varepsilon_{xx}^{0}U.$$

It is known from Equation (2) that the CTE and CEE are independent of the beam curvature and strain under the second-order approximation, and vice versa. Therefore, we can obtain the CTE and the CEE from a tube’s calculated or experimental data. Although existing atomic calculations have shown that the elastic constants of SWCNTs decrease slightly with a temperature increase [30], this effect is ignored in the second-order approximation. Similarly, quantum computing shows that the electrostatic field affects elastic constants, but its mechanism has not been studied in detail [22,23]. In the present paper, we restrict the applied electrostatic field, U<0.01 V/nm, so the electrostatic field’s influence on two stiffnesses is ignored [23]. For a slender beam, the curvature and axial strain are [31]
(3)$$\kappa \approx \frac{\partial^{2}w}{\partial x^{2}}\left[1-\frac{3}{2}{\left(\frac{\partial w}{\partial x}\right)}^{2}\right],$$
(4)$$\varepsilon_{xx}^{0}=\frac{\partial u}{\partial x}+\frac{1}{2}{\left(\frac{\partial w}{\partial x}\right)}^{2}-\frac{1}{2}{\left(\frac{\partial u}{\partial x}\right)}^{2}-\frac{1}{2}\frac{\partial u}{\partial x}{\left(\frac{\partial w}{\partial x}\right)}^{2}-\frac{1}{8}{\left(\frac{\partial w}{\partial x}\right)}^{4}.$$

Here, u and w are the displacements in directions of the x- and y-axes of the SWCNT’s centroid locus. The dynamic energy density of the tube is
(5)$$D=\frac{1}{2}m\left[{\left(\frac{\partial u}{\partial t}\right)}^{2}+{\left(\frac{\partial w}{\partial t}\right)}^{2}\right].$$

Here, m is the mass per unit length, l is the SWCNT’s length, and we ignore the kinetic energy induced by cross-section rotations. The energy density due to external force is
(6)$$W=\bar{f}\left(x,t\right)w+\bar{g}\left(x,t\right)u,$$

Here, $\bar{f}$ and $\bar{g}$ are external forces in the directions of the y- and x-axes. Therefore, the structure’s Lagrangian is $H=\int_{t_{0}}^{t_{1}}\int_{0}^{l}\left(T-F-W\right)dxdt$. Let δH=0 (Hamilton principle [32]), and the motion equations are
(7)$$m\frac{\partial^{2}u}{\partial t^{2}}-k_{S}\left(1-k_{U}U-Tk_{T}\right)\frac{\partial }{\partial x}\left[\frac{\partial u}{\partial x}+\frac{1}{2}{\left(\frac{\partial w}{\partial x}\right)}^{2}\right]=\bar{g}\left(x,t\right)$$
(8)$$\begin{array}{l} m\frac{\partial^{2}w}{\partial t^{2}}+k_{S}\left(k_{U}U+Tk_{T}\right)\frac{\partial^{2}w}{\partial x^{2}}+k_{B}\frac{\partial^{4}w}{\partial x^{4}}-k_{S}\left(1+k_{U}U+Tk_{T}\right)\frac{\partial }{\partial x}\{\frac{\partial w}{\partial x}[\frac{\partial u}{\partial x}\\ +\frac{1}{2}{\left(\frac{\partial w}{\partial x}\right)}^{2}]\}-k_{B}\left[12\frac{\partial w}{\partial x}\frac{\partial^{2}w}{\partial x^{2}}\frac{\partial^{3}w}{\partial x^{3}}+3{\left(\frac{\partial w}{\partial x}\right)}^{2}\frac{\partial^{4}w}{\partial x^{3}}+3{\left(\frac{\partial^{2}w}{\partial x^{2}}\right)}^{3}\right]=\bar{f}\left(x,t\right) \end{array}$$

Equations (7) and (8) are the SWCNT’s plane motion equations with the independent extensional and bending stiffness for the two immovable ends. For the tubes with hinged support at both ends, the boundary conditions are
(9)$$u=w=\frac{\partial^{2}w}{\partial x^{2}}=0,\;at\;x=0,l.$$

Other boundary conditions with two immovable ends are consistent with the classical beams. For slender beams, the longitudinal displacement is mainly induced by lateral deformation, and the longitudinal inertial forces can be ignored [31,32]. Thus, Equation (7) is simplified as $\partial \left[\partial u/\partial x+{\left(\partial w/\partial x\right)}^{2}/2\right]/\partial x=0$. For $\bar{g}\left(x,t\right)=0$, integrating this equation as
(10)$$\frac{\partial u}{\partial x}=-\frac{1}{2}{\left(\frac{\partial w}{\partial s}\right)}^{2}+c_{1},\;u=-\frac{1}{2}\int_{0}^{x}{\left(\frac{\partial w}{\partial s}\right)}^{2}ds+c_{1}x+c_{2}.$$

For boundary conditions of Equation (9), $c_{1}=\frac{1}{2l}\int_{0}^{l}{\left(\partial w/\partial x\right)}^{2}dx$ and $c_{2}=0$ in Equation (10) [30]. This further obtains
(11)$$\frac{\partial u}{\partial x}=-\frac{1}{2}{\left(\frac{\partial w}{\partial s}\right)}^{2}+\frac{1}{2l}\int_{0}^{l}{\left(\frac{\partial w}{\partial x}\right)}^{2}dx.$$

Substituting Equation (11) into Equation (8) produces
(12)$$\begin{array}{l} m\frac{\partial^{2}w}{\partial t^{2}}+C\frac{\partial w}{\partial t}+k_{S}\left(k_{U}U+Tk_{T}\right)\frac{\partial^{2}w}{\partial x^{2}}+k_{B}\frac{\partial^{4}w}{\partial x^{4}}-\frac{k_{S}\left(1+k_{U}U+Tk_{T}\right)}{2l}\frac{\partial^{2}w}{\partial x^{2}}\\ \int_{0}^{l}{\left(\frac{\partial w}{\partial x}\right)}^{2}dx-k_{B}\left[12\frac{\partial w}{\partial x}\frac{\partial^{2}w}{\partial x^{2}}\frac{\partial^{3}w}{\partial x^{3}}+3{\left(\frac{\partial w}{\partial x}\right)}^{2}\frac{\partial^{4}w}{\partial x^{4}}+3{\left(\frac{\partial^{2}w}{\partial x^{2}}\right)}^{3}\right]=\bar{f}\left(x,t\right). \end{array}$$
where we add a viscous damping term, c(∂w/∂x), to Equation (12). For the statics problem, Equation (12) is simplified as
(13)$$\begin{array}{l} k_{B}\frac{\partial^{4}w}{\partial x^{4}}+k_{S}\left(k_{U}U+Tk_{T}\right)\frac{\partial^{2}w}{\partial x^{2}}-\frac{k_{S}\left(1+k_{U}U+Tk_{T}\right)}{2l}\frac{\partial^{2}w}{\partial x^{2}}\int_{0}^{l}{\left(\frac{\partial w}{\partial x}\right)}^{2}dx\\ -k_{B}\left[12\frac{\partial w}{\partial x}\frac{\partial^{2}w}{\partial x^{2}}\frac{\partial^{3}w}{\partial x^{3}}+3{\left(\frac{\partial w}{\partial x}\right)}^{2}\frac{\partial^{4}w}{\partial x^{4}}+3{\left(\frac{\partial^{2}w}{\partial x^{2}}\right)}^{3}\right]=\bar{f}\left(x\right). \end{array}$$

For the tube hinged at both ends, the boundary conditions of w in Equations (12) and (13) are consistent with Equation (9). Using MD calculations, the extensional and bending stiffnesses are obtained as $k_{S}=\alpha \left(d-d_{0}\right)$ and $k_{B}=\beta {\left(d-d_{0}\right)}^{3}$. Here, α=1128.15 nN/nm, β=142.54 nN/nm, and $d_{0}=2.7\times 10^{-7}\;nm$ are independent fitting parameters [3,11]. Because $d_{0}$ is much smaller than the tube’s diameter d, we let $d_{0}=0$, as shown in Ref. [11]. In the classical beam model, the axial extensional stiffness $k_{S}=Eh\pi d$ and the bending stiffness $k_{B}=Eh\pi \left(d^{3}+dh^{2}\right)/8$ [32]. The MD calculations display E=1.086 TPa for h=0.335 nm [11]. In the present paper, we take an armchair (5,5)SWCNT hinged at two ends to study the influence of electrostatic field and temperature on the mechanical properties. Thus, its diameter is d=0.678 nm. Other physical and geometric parameters are shown in Figure 1 and Table 1. Table 1 shows that the bending stiffness of the classical model is much greater than that of the independent stiffness model. The classical beam theory has $k_{B}/k_{S}=\left(d^{2}+h^{2}\right)/8$, while the independent stiffness model has $k_{B}/k_{S}=\beta \alpha^{-1}d^{2}\approx 0.0575d^{2}$. The ratio of independent stiffness theory is smaller than that of classical theory. The CTE in Table 1 comes from Ref. [20], and the CEE is $k_{U}=0.025\;nm/V$, which is obtained by the linear fitting of the FDT calculations of (3,3)SWCNT in Ref. [22].

## 3. Example and Discussion

### 3.1. Static Bending Deformation and Buckling

In this section, an armchair (5,5)SWCNT is taken as an example to discover the temperature and electrostatic field effects on the mechanical properties. Since Equations (12) and (13) are the nonlinear differential
integral
equation, and it is difficult to obtain accurate analytical solutions of these two equations, we use the Galerkin method to obtain approximate analytical solutions [32]. Suppose the solution of Equation (12) is
(14)$$w=\sum_{j=1}^{n}{\hat{\eta }}_{j}\left(t\right)\sin\left(\frac{j\pi x}{l}\right).$$

Substituting Equation (14) into Equation (12) here take only the first term and lets ${\hat{\eta }}_{1}=\eta$ for the sake of simplification, multiplies it by sin(πx/l) on two sides, and then, using the integrals in [0,l] (Galerkin integral [31,32]), we obtain
(15)$$\frac{ml}{2}\ddot{\eta }+\frac{\hat{c}l}{2}\dot{\eta }+k_{1}\eta +k_{2}\eta^{3}=2l\pi^{-1}\bar{f}.$$

In the following sections, we shall denote the derivative with respect to time by placing a dot above the letter. In Equation (15), suppose the transverse load is a uniform load, namely $\bar{f}\left(x,t\right)=\bar{f}\left(t\right)$. The parameters in Equation (15) are
(16)$$k_{1}=\frac{\pi^{4}k_{B}}{2l^{3}}-\frac{\pi^{2}k_{S}\left(k_{U}U+Tk_{T}\right)}{2l},\;k_{2}=\frac{\pi^{4}k_{S}\left(1+k_{U}U+Tk_{T}\right)}{8l^{3}}-\frac{3\pi^{6}k_{B}}{4l^{5}}.$$

By omitting the inertial and damping terms, the equation of the static problem is obtained as
(17)$$k_{1}\eta +k_{2}\eta^{3}=\frac{2l}{\pi }\bar{f}$$

In fact, Equation (17) can be obtained by applying the Galerkin method to Equation (13). The relationship between bending deformations and applied loads can be obtained by Equation (17), as shown in Figure 2. The figure shows that the classical model’s deformations are less than those of the independent stiffness model under the same conditions because the classical beam theory overestimates the bending stiffness of SWCNTs. If the transverse load $\bar{f}=0$ and the temperature or electrostatic field intensity exceeds its critical value, it is known from the expression of $k_{1}$ that the SWCNTs will buckle. The critical temperature and critical electric field intensity are determined as in classical beam theory,
(18)$$k_{C}U+Tk_{T}=\frac{\pi^{2}k_{B}}{l^{2}k_{S}}.$$

The relationship between critical temperature, critical electric field intensity, and critical tube length can be obtained by Equation (18), as shown in Figure 3, Figure 4 and Figure 5. These figures clearly demonstrate that the classical beam model’s critical values are much higher than those of the independent stiffness model. For example, the classical model’s critical buckling temperature is T=1140 K, while that of the independent stiffness theory is T=910 K for U=0.01 V/nm, as shown in Figure 5. The difference is 230 K, which is dramatic in practice. The buckling of SWCNTs will create initial curvatures, which can induce quadratic nonlinearity terms in the model and make the mechanical response appear significantly different from that of buckling-free SWCNTs [33,34]. This issue may require intensive research. From Equation (18), it can also be found that the buckling is more sensitive to the electric field than the temperature due to $k_{C}\gg k_{T}$. This issue has not yet received attention so far.

### 3.2. Nonlinear Bending Vibrations

This section focuses on the effects of temperature and electric field on the nonlinear vibrations of SWCNTs. Because the vibrational frequencies of SWCNTs are extremely high, it is convenient to introduce dimensionless variables $\bar{\eta }=\eta /l$ and $\bar{t}=t/\omega_{0}$. Here, $\omega_{0}=\sqrt{\pi^{4}k_{B}/ml^{4}}$ is nature frequency for (T,U)=(0,0). Thus, Equation (15) is written as
(19)$$\ddot{\bar{\eta }}+2\bar{c}\dot{\bar{\eta }}+{\bar{k}}_{1}\bar{\eta }+{\bar{k}}_{2}{\bar{\eta }}^{3}=\hat{f},$$
where
(20)$$\begin{array}{l} \bar{c}=\frac{\hat{c}}{2m\omega_{0}},\;k_{1}=1-\frac{\pi^{2}k_{S}\left(k_{U}U+Tk_{T}\right)}{ml^{2}\omega_{0}^{2}},\\ k_{2}=\frac{\pi^{4}k_{S}\left(1+k_{U}U+Tk_{T}\right)}{4ml^{2}\omega_{0}^{2}}-\frac{3\pi^{6}k_{B}}{2ml^{4}\omega_{0}^{2}},\;\hat{f}=\frac{4\bar{f}}{\pi ml\omega_{0}^{2}}. \end{array}$$

To research the oscillations, we perturb Equation (19) by letting $\bar{c}=2\varepsilon^{2}c$ and $\varepsilon^{3}\hat{f}=f$. ε is a small parameter, and ε=0.1 is used in this paper. Consequently, Equation (19) can be rewritten as
(21)$$\ddot{\bar{\eta }}+\omega_{1}^{2}\bar{\eta }+2\varepsilon^{2}c\dot{\bar{\eta }}+k_{3}{\bar{\eta }}^{3}=\varepsilon^{3}f\cos\omega \;\bar{t},$$

Here, $\omega_{1}^{2}=k_{1}$. The method of multiple scales, a classical perturbation method used to solve the weak nonlinear differential equation [35], is used to solve Equation (21). The solution of Equation (21) can be represented by an expansion of $\bar{\eta }$ as
(22)$$\bar{\eta }=\varepsilon \eta_{0}\left(T_{0},T_{2}\right)+\varepsilon^{3}\eta_{1}\left(T_{0},T_{2}\right),$$

Here, $T_{0}=\bar{t}$ and $T_{2}=\varepsilon^{2}\bar{t}$. Substituting Equation (22) into (21), and equating the coefficients of ε and $\varepsilon^{3}$ on both sides, gives [36]:(23)$$D_{0}^{2}\eta_{0}+\omega_{1}^{2}\eta_{0}=0,$$
(24)$$D_{0}^{2}\eta_{1}+\omega_{1}^{2}\eta_{1}=-2D_{0}D_{2}\eta_{0}-2cD_{0}\eta_{0}-k_{3}\eta_{0}^{3}+f\left(\bar{x}\right)\cos\left(k_{1}T_{0}+\sigma T_{2}\right),$$

Here, $\omega -\omega_{1}=\varepsilon^{2}\sigma$. $D_{0}$ and $D_{2}$ signify derivatives of $T_{0}$ and $T_{2}$, respectively. According to the ordinary differential equation theory, the solution of Equation (23) is
(25)$$\eta_{0}=A\left(T_{2}\right)\exp\left(i\omega_{\;1}T_{0}\right)+cc,\;$$

Here, *cc* denotes the complex conjugate of the preceding term. Substituting $\eta_{0}$ into Equation (24) gives
(26)$$\begin{array}{l} D_{0}^{2}\eta_{1}+\omega_{1}^{2}\eta_{1}=-\left[2i\omega_{\;1}\left(D_{2}A+cA\right)+3k_{3}A^{2}\bar{A}\right]\exp\left(i\omega_{\;1}T_{0}\right)\\ -k_{3}A^{3}\exp\left(3i\omega_{\;1}T_{0}\right)+\frac{1}{2}f\exp\left[i\left(\omega_{\;1}T_{0}+\sigma T_{2}\right)\right]+cc. \end{array}$$

Eliminating secular terms [36] from Equation (26) gives
(27)$$2i\omega_{\;1}\left(D_{2}A+cA\right)+3k_{3}A^{2}\bar{A}-\frac{1}{2}f\exp\left(i\sigma T_{2}\right)=0.$$

Let A=αexp(iβ)/2; here, α and β are real functions of $T_{2}$. Then, separating the real and imaginary parts of Equation (27) gives
(28)$$D_{2}\alpha =-c\alpha +\frac{f}{2\omega_{\;1}}\sin\left(\alpha T_{2}-\beta \right),\;$$
(29)$$\alpha D_{2}\beta =\frac{3k_{3}}{8\omega_{\;1}}\alpha^{3}-\frac{f}{2\omega_{\;1}}\cos\left(\alpha T_{2}-\beta \right).\;$$

A steady-state motion will occur when $D_{2}\alpha =D_{2}\beta =0$. In addition, α, β can be obtained through the following nonlinear equations:(30)$$-c\alpha +\frac{1}{2}\frac{f}{\omega_{\;1}}\sin\gamma =0,\;\sigma \;\alpha -\frac{3}{8}\frac{k_{3}}{\omega_{\;1}}\alpha^{3}+\frac{1}{2}\frac{f}{\omega_{\;1}}\cos\gamma =0,$$

Here, $\gamma =\alpha T_{2}-\beta$. Equation (30) gives
(31)$$\left[c^{2}+{\left(\sigma -\frac{3k_{2}}{8\omega_{\;1}}\alpha^{2}\right)}^{2}\right]\alpha^{2}=\frac{f^{2}}{4\omega_{1}^{2}}.$$

Equation (31) is an implicit equation for the response amplitude, α, as a function of the detuning parameter σ and the excitation amplitude f. Substituting α and γ into Equation (19) gives a first-order approximate solution of Equation (21):(32)$$\bar{\eta }\approx \varepsilon \eta_{0}=\varepsilon \;\alpha \cos\left(\omega \;\bar{t}-\gamma \right).\;$$

The damping coefficient of SWCNTs has not yet been researched intensively, so $\bar{c}=0.01$ is used here. Other physical and geometric parameters are l=10 nm, $m=1.619\times 10^{-15}\mathrm{kg}/m$, and E=1.086 TPa [11]. Thus, $\omega_{0}^{2}=3.423\times 10^{23}\;s^{-2}$ for the classical model, and $\omega_{0}^{2}=2.642\times 10^{23}\;s^{-2}$ for the independent stiffness model. From Equation (31), we obtain a function of vibrational amplitude as other parameters, as shown in Figure 6, Figure 7 and Figure 8. These figures reveal three aspects of the main vibrational features: first, since the bending stiffness of the classical model is greater than that of the independent stiffness model, the vibrational amplitudes of the classical model are less than those of the independent stiffness model with small loads, but the order of the two amplitudes will reverse with the increase of the loads, as shown in Figure 6; second, when the vibrational amplitudes appear as multi-values due to bifurcation, the amplitudes will jump and significantly affect the motion of SWCNTs, as shown in Figure 7 and Figure 8; third, the electric field intensity and temperature significantly affect the bifurcation positions and vibrational amplitudes. In fact, the load bifurcation positions of the classical model are greater than those of the independent stiffness model, as shown in Figure 7 and Figure 8. For example, the load f=0.95 exceeds the independent stiffness model’s bifurcation point for (c,σ,U,T)=(1, 6, 0.01, 0), so the two vibrational amplitudes are identical for the independent stiffness model with different initial values. However, the difference in the classical model’s amplitudes is more than three times for different initial values, as shown in Figure 9 and Figure 10. Because bifurcation points are sensitive to temperature and electric field intensity, the classical beam model may no longer be suitable for the accurate analysis of SWCNTs’ nonlinear oscillations.

It is necessary to note that the theory’s validity needs to be checked by experiments. However, the bending mechanical properties of SWCNTs under combined physical fields have not yet attracted researchers’ attention. Therefore, we do not find any calculated and experimental data to compare with the present theoretical results. For example, the independent extensional stiffness and bending stiffness are obtained by MD calculations in Refs. [3,11], but the two papers do not provide data of deformations. The experiments of SWCNT bending deformations usually use long tubes (>100 nm), but the bending stiffness can be ignored for these. Researchers have not realized that temperature and electrostatic field significantly affect the bending mechanical properties of SWCNTs. These issues call for further studies, both experimental and theoretical.

## 4. Conclusions

In the present paper, a new thermal–electro-mechanical coupling beam model of SWCNTs, in which the bending stiffness and extensional stiffness are independent, is proposed. The static bending deformations, buckling, and nonlinear forced vibrations are researched through the new model and the classical beam model. The results show that the classical model significantly underestimates the influence of temperature and electrostatic field on mechanical properties because it overestimates the bending stiffness of SWCNTs. These influences are reflected in three main aspects: first, the independent stiffness model has greater static bending deformations than the classical beam model under the same conditions; second, the independent stiffness model has a lower critical buckling temperature and critical buckling electric field intensity than the classical beam model; third, for nonlinear vibrations, the independent stiffness model has smaller bifurcation loads than the classical beam model, and the bifurcation loads sensitively depend on the temperature and the electric field intensity. The present research also shows that for a precise understanding of the mechanical properties of SWCNTs, independent stiffness, temperature and electric field intensity should be considered.

## Figures and Tables

**Figure 1 nanomaterials-11-00923-f001:**
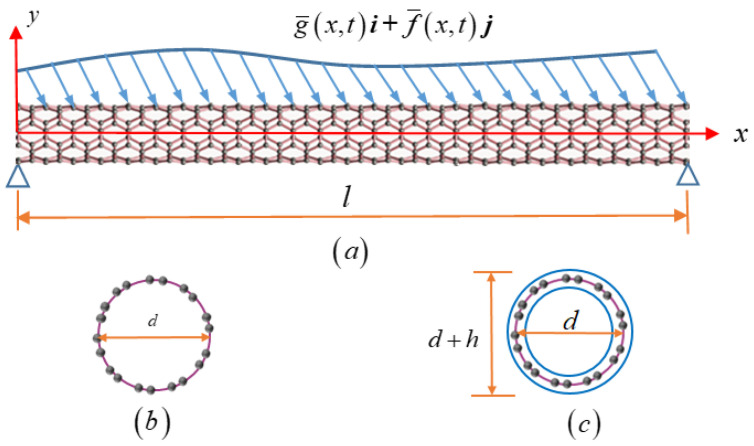
Schematic configuration of a single-walled carbon nanotube (SWCNT). (**a**) Front elevation, (**b**) Plan, (**c**) Equivalent cross-section.

**Figure 2 nanomaterials-11-00923-f002:**
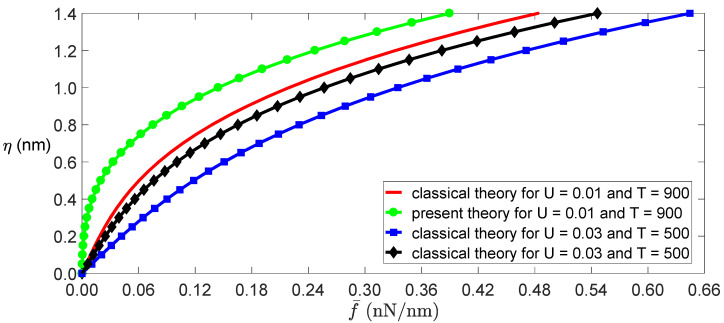
Amplitude of the deformation as a function of the load for the given temperatures and electric fields.

**Figure 3 nanomaterials-11-00923-f003:**
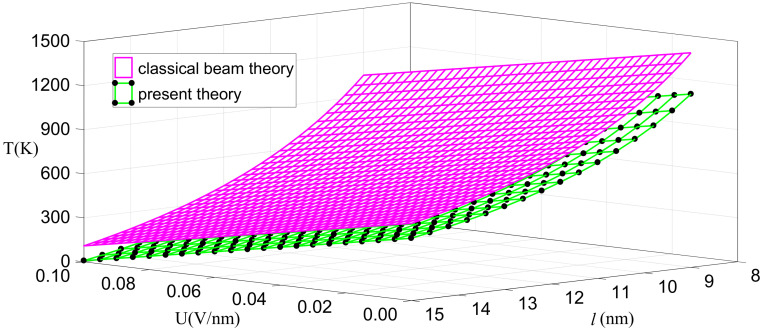
Critical temperature as a function of critical electric field intensity and critical length.

**Figure 4 nanomaterials-11-00923-f004:**
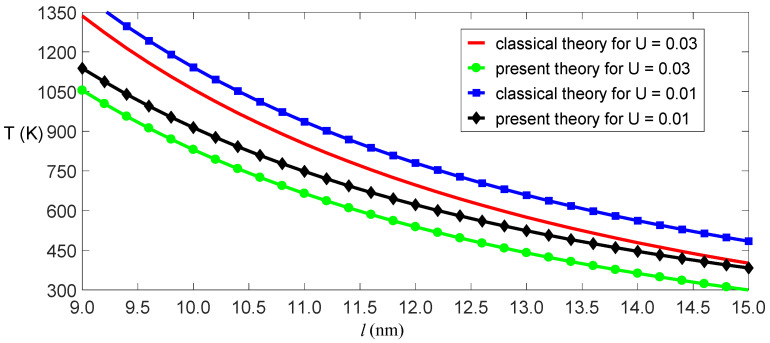
Critical temperature as a function of critical length for the given critical electric fields.

**Figure 5 nanomaterials-11-00923-f005:**
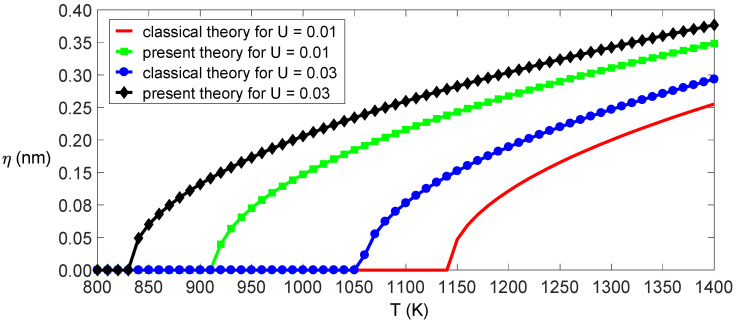
Amplitude of deformation as a function of temperature for the given electric fields.

**Figure 6 nanomaterials-11-00923-f006:**
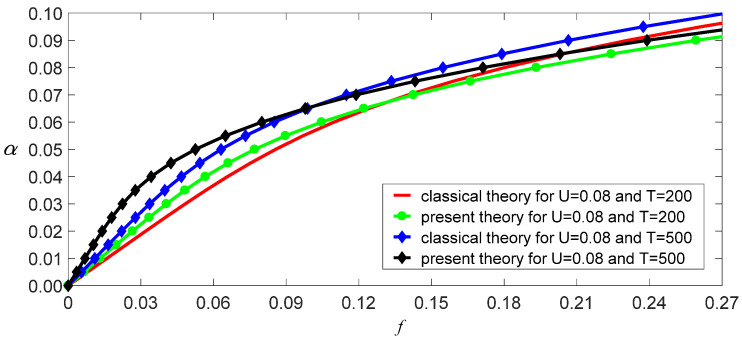
Load–response curves of vibration amplitudes with two temperatures for (c,σ,U)=(1, 0, 0.08).

**Figure 7 nanomaterials-11-00923-f007:**
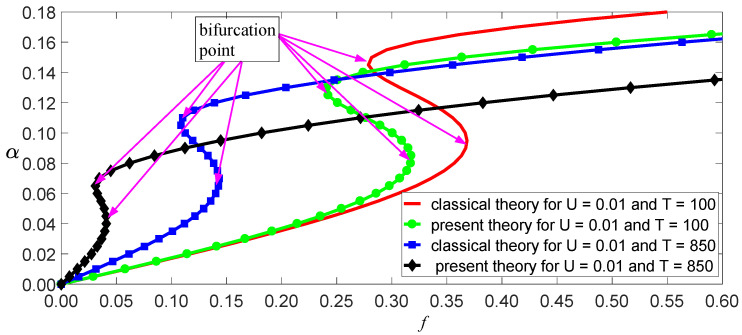
Load–response curves of vibration amplitudes with two temperatures for (c,σ,U)=(1, 3, 0.01).

**Figure 8 nanomaterials-11-00923-f008:**
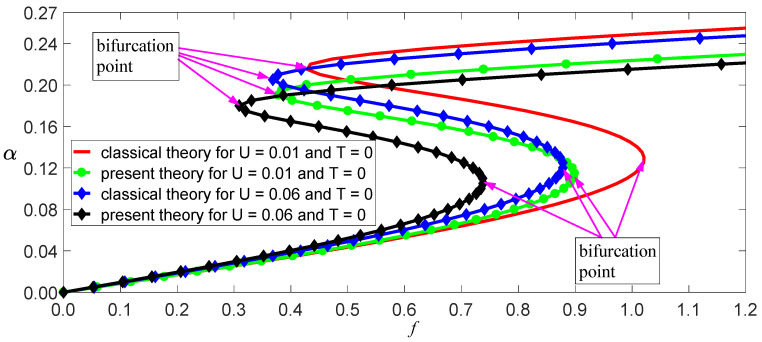
Load–response curves of vibration amplitudes with two electric fields for (c,σ,T)=(1, 6, 0).

**Figure 9 nanomaterials-11-00923-f009:**
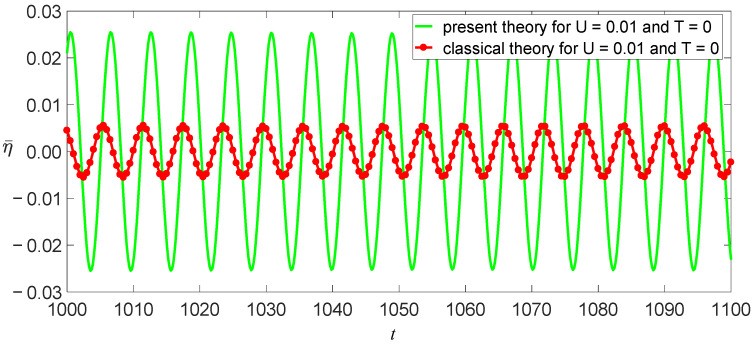
Two models’ time evolutions of η for (c,σ,f)=(1, 6, 0.95) at the initial value $\left(\eta_{0},{\dot{\eta }}_{0}\right)=\left(0.012,\;0\right)$.

**Figure 10 nanomaterials-11-00923-f010:**
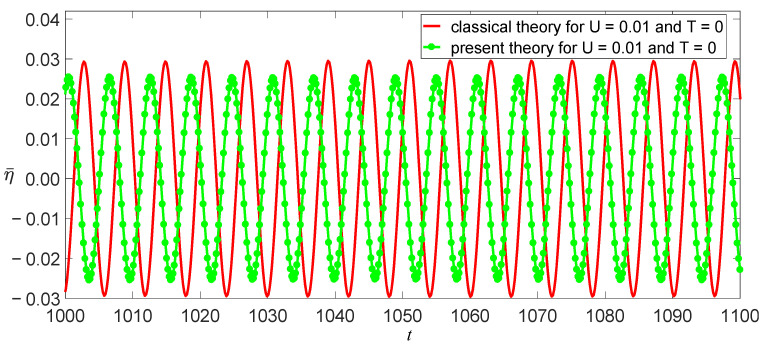
Two models’ time evolutions of η for (c,σ,f)=(1, 6, 0.95) at the initial value $\left(\eta_{0},{\dot{\eta }}_{0}\right)=\left(0.025,\;0\right)$.

**Table 1 nanomaterials-11-00923-t001:** Physical parameters for chiral (5,5) SWCNT (here, α, β and E come from Ref. [3]).

	$k_{S},\left(nN/nm\right)$	$k_{B},\left(nN\cdot nm^{2}\right)$	$k_{T},\left(T^{-1}\right)$	$k_{U},\left(nm/V\right)$
Classical model	$0.786\times 10^{3}$	$0.057\times 10^{3}$	$6.0\times 10^{-6}$	0.025
Independent stiffness model	$0.765\times 10^{3}$	$0.044\times 10^{3}$	$6.0\times 10^{-6}$	0.025

## Data Availability

The data presented in this study are available on request from the corresponding author.

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
