# Peer review of "Beam Theory of Thermal–Electro-Mechanical Coupling for Single-Wall Carbon Nanotubes"

_nanomaterials, 2021, doi:10.3390/nano11040923_

Round 1

Reviewer 1 Report

The paper is clear and sound, it shows how independent approach to axial and bending stiffnesses influence the resulting calculated mechanical properties of the single-walled nanotubes. However, it is unclear if the new approach makes the theory more adequate since there is no any reference to experimental data, which would support the theoretical conclusions. It harms a lot and makes the overall merit low. The Reviewer believes that the paper cannot be published in the present form and requires major review concerning link between calculated and experimental data.

It is also recommended that the authors refine their English due to many awkward expressions used through the text. Also, it is necessary to correct a misprint in the word "theory" (spelled as "thoery" so many times in the legends to the Figures).

Author Response

Comment: The paper is clear and sound, it shows how independent approach to axial and bending stiffnesses influence the resulting calculated mechanical properties of the single-walled nanotubes. However, it is unclear if the new approach makes the theory more adequate since there is no any reference to experimental data, which would support the theoretical conclusions. It harms a lot and makes the overall merit low. The Reviewer believes that the paper cannot be published in the present form and requires major review concerning link between calculated and experimental data.

It is also recommended that the authors refine their English due to many awkward expressions used through the text. Also, it is necessary to correct a misprint in the word "theory" (spelled as "thoery" so many times in the legends to the Figures).

Reply: We thank the reviewers for the recommendations. We carefully proofread the manuscript to minimize typographical, grammatical, and bibliographical errors.

The theory’s validity needs to be checked by experiment. However, the mechanical properties of SWCNTs under combined physical fields have not attracted the researchers’ attention so far. There are no calculated and experimental data for an SWCNT’s thermal-electro-mechanical coupling properties in the existing literature. Therefore, we cannot find any data to compare the present theoretical results. For example, the independent extensional stiffness and bending stiffness are obtained by MD calculations in Ref. [3] and [11], but the two papers did not give the data of deformations for loads. The SWCNTs experiments usually used large length tubes (>100 nm), where the bending stiffness can be ignored. So these data also cannot be used to check the results in this paper. In fact, researchers have not realized that temperature and electrostatic field significantly affect the bending mechanical properties. Our paper theoretically predicted the impact of the three factors for the first time. This may attract researchers to pay attention to this critical issue. In the revision, we have explained this issue at the end of Section 1 and Section 3 and write them in red.

Reviewer 2 Report

It has been widely recognized that carbon nanotubes have attractive mechanical properties and thus many applications such as light-weight cable and nanotube radio have been studied. In this paper, a new model describing the mechanical properties of carbon nanotubes is proposed. The effects of temperature and electrostatic field can be included by the model and the results suggest that both substantially affect the mechanical properties of carbon nanotubes. The proposed mechanism seems to be sound and very useful for nanotube community. Hence, I recommend its publication in Nanomaterials. My only concern is that it would be better if the authors compare their predictions with the previous experimental data and discuss it.

Author Response

Comment: It has been widely recognized that carbon nanotubes have attractive mechanical properties and thus many applications such as light-weight cable and nanotube radio have been studied. In this paper, a new model describing the mechanical properties of carbon nanotubes is proposed. The effects of temperature and electrostatic field can be included by the model and the results suggest that both substantially affect the mechanical properties of carbon nanotubes. The proposed mechanism seems to be sound and very useful for nanotube community. Hence, I recommend its publication in Nanomaterials. My only concern is that it would be better if the authors compare their predictions with the previous experimental data and discuss it.

Reply: We thank the reviewers for the recommendations. We carefully proofread the manuscript to minimize typographical, grammatical, and bibliographical errors. At the time of this paper writing, there is no literature to report the effects of independent bending and extensional stiffness on the vibrations of SWCNTs. Due to the experimental difficulty, there is no bending deformational experimental data for short SWCNTs (<30 nm) in the existing literature. The properties of an SWCNT’s thermal-electro-mechanical coupling have not been paid attention to by researchers so far. Our paper theoretically predicted the impact of the three factors for the first time. Therefore, we cannot find any data to compare the present theoretical results in the existing literature. This paper may attract researchers to pay attention to this important issue. In the revision, we have explained this issue at the ends of Section 1 and Section 3, and write them in red.

Round 2

Reviewer 1 Report

The authors have clarified all the dubious entries. The manuscript in the present form can be published.